# Characteristics of "chūnibyō" identified by a questionnaire

**Masafumi Shimoda** \*, **Kozo Morimoto, Yoshiaki Tanaka, Kozo Yoshimori, Ken Ohta**

Respiratory Disease Center, Fukujuji Hospital, Japan Anti-Tuberculosis Association (JATA), Kiyose City, Tokyo, Japan

\* shimodam@fukujuji.org

## Abstract

### Background

"Chūnibyō" is a term that represents a distinctive, transient mental state during puberty in Japan, but its characteristics and precise definition have not been standardized. Increased awareness of chūnibyō could lead to a better environment for those who experience it. This study aimed to identify the characteristics of and problems related to chūnibyō using an anonymous questionnaire.

### Materials and methods

An anonymous online questionnaire was conducted in February 2021 in Japan. In total, 314 volunteers completed the anonymous online questionnaire. Respondents were divided into the chūnibyō group (n = 122) and the non-chūnibyō group (n = 192), and the questionnaire responses were compared between the two groups. Furthermore, the responses were compared between the chūnibyō with problems subgroup (n = 82) and the other subgroup (n = 232). The main outcome was the identification of the chūnibyō group based on the responses to the item "I have experienced chūnibyō" or "I have been told that I exhibited chūnibyō".

### Results

The median age of the chūnibyō group was 31 years old; this group was predominantly male (n = 79, 64.8%) and had a relatively high proportion of respondents with any problems (n = 82, 67.2%). The chūnibyō group had higher proportions of respondents who felt that academic tests did not reflect their true worth (n = 58 (47.5%) vs. n = 66 (35.4%), p = 0.024), who felt uncomfortable in the world (n = 77 (61.1%) vs. n = 67 (34.9%), p<0.001), and who had an imaginary/fantasy friend or boyfriend/girlfriend (n = 39 (32.0%) vs. n = 10 (5.2%), p<0.001). The results were similar between the chūnibyō with problems subgroup and the other subgroup. Eighty respondents (25.4%) had negative impressions of chūnibyō, whereas twenty-one respondents (6.7%) had positive impressions.

**Data Availability Statement:** We uploaded data availability statement of our study as supplemental file. Furthermore, we will provide the available data if a request is sent to "shimodam@fukujuji.org".

**Funding:** The authors received no specific funding for this work.

**Competing interests:** The authors have declared that no competing interests exist.

**Abbreviations:** AUC, the area under the receiver operating characteristic curve; CI, confidence interval; ICD-11, International Classification of Diseases 11th Revision; IQR, interquartile range; Q, question; ROC, receiver operating characteristic; SNS, social networking service.

## Conclusions

This study is the first to report the characteristics of chūnibyō by collecting the experiences and thoughts of people who experienced chūnibyō.

## Introduction

Chūnibyō" is a term that was coined to represent a distinctive, transient mental state that can occur during puberty in Japan [1–4]. The concept of chūnibyō was developed on a radio program in the 1990s and spread rapidly during the 2000s [5]. "Chū-ni" means the eighth-grade period, and "byō" means a disease; therefore, chūnibyō is translated into "eighth-grader syndrome" [6] or "eighth-grade disease" [1]. However, chūnibyō is not a disease classified in the International Classification of Diseases 11th Revision (ICD-11) [7]. The behaviours that are characteristic of chūnibyō, such as attempts to make oneself look more important, can be observed during pubertal emotional development [1,5,8,9], and the behaviours associated with chūnibyō are generally resolved spontaneously after puberty [4,5]. Those behaviours are different from usual changes experienced during puberty. People with chūnibyō act aggressive, defiant, delusional, withdrawn, boastful, and/or in ways that refer to characters specific to Japanese culture, such as those found in anime, manga, cosplay, and gaming, which draw on teenage fantasies about temporarily possessing supernatural powers during puberty [1,5,6]. For example, people with chūnibyō sometimes apply a bandage to an uninjured arm to "seal in" the power granted them by a devil, insist that they can see a supernatural being, and call themselves elevated names such as "Researcher" and "Dream Soldier" [1,4]. Some people experience the following problems associated with chūnibyō: poor communication skills, stress without a specific cause, bullying, remaining chūnibyō even as an adult, and others [1,2,4,5]. Since chūnibyō is not disease, interest in chūnibyō is low, and sufficient countermeasures have not been taken for children with chūnibyō. If people are more aware of chūnibyō, families, teachers, friends, and adults around children with chūnibyō might be able to provide a better environment. However, no study has reported the characteristics of chūnibyō in detail, and the definition of chūnibyō has not been standardized. Therefore, we conducted an anonymous questionnaire regarding chūnibyō and identified the characteristics of chūnibyō by investigating the problems experienced due to chūnibyō.

## Materials and methods

### Study design and setting

This study was conducted with 337 adult volunteers (age≥20 years old) in February 2021. All respondents replied to an anonymous online questionnaire in Japanese, which is translated into English in Table 1. Respondents were recruited via the Fukujuji Hospital Respiratory Disease Center official blog; social networking services (SNSs) such as Twitter, Facebook, Instagram, and LINE; and internet bulletin boards. Respondents were divided into a chūnibyō group (n = 122) and a non-chūnibyō group (n = 192), and the answers to the questions were compared between the two groups. Furthermore, the answers to the questions were compared between a chūnibyō with problems subgroup (n = 82) and the others group (n = 232). Ten respondents who did not provide their informed consent and thirteen respondents who had been diagnosed with a psychiatric disease in middle/high school were excluded. The flowchart of the study is shown in Fig 1. Informed consent was obtained from all respondents, although

**Table 1. The English translation of the questionnaire used in the study.**

| Question A | Answer |
| --- | --- |
| QA-1. What is your age? | Age |
| QA-2. What is your gender? | • male<br>• female |
| QA-3. Did you experience chūnibyō during your middle/high school years? | • Yes<br>• No<br>• I do not think I did, but I was told I did.<br>• I am not sure if I did, and nobody told me I did. |
| QA-4. Have you had any problems due to chūnibyō or any other problems related to puberty? (Multiple answers are allowed) | • No nor have I experienced chūnibyō.<br>• I had difficulties communicating.<br>• Relationships with family and/or friends deteriorated.<br>• I felt stressed with no clear cause.<br>• I was picked on for experiencing chūnibyō.<br>• I was bullied.<br>• I became socially withdrawn and/or unemployed.<br>• I still experience chūnibyō now.<br>• Other |
| QA-5. Do you have any embarrassing experiences you would like to erase? Is it a bad memory? | • Yes, it was both embarrassing and bad.<br>• Yes, it was embarrassing, but the memory itself is not bad.<br>• No |
| QA-6. Did you have any diseases that required a visit to the hospital? | • No<br>• Internal disease<br>• Surgery<br>• Dermatological, otorhinolaryngologic, urological, or ophthalmic disease<br>• Psychiatric disorder<br>• Other |
| Question B: Please answer the questions below while thinking back on your chūnibyō (or middle/high school) days. | |
| QB-1. Did you ever think that academic tests were not enough to measure your true abilities? | • Yes<br>• No |
| QB-2. Did you ever feel that something was wrong with the world? | • Yes<br>• No |
| QB-3. Did you call yourself something other than "Boku"/"Ore" if male or "Watasi"/"Atasi" if female? | • Yes<br>• No |
| QB-4. Did you have any imaginary/fantasy friends or boyfriends/girlfriends? | • Yes<br>• No |
| QB-5. Have you ever confessed your love to others, or have others ever confessed their love to you when you were middle/high school students? | • Yes<br>• No |
| QB-6. If someone had told you that you had chūnibyō when you were an adolescent, how would you have reacted? (Multiple answers are allowed) | • Accepted it<br>• Rejected it<br>• Felt good about it.<br>• Had a negative reaction. |

All respondents replied to this anonymous online questionnaire in Japanese.

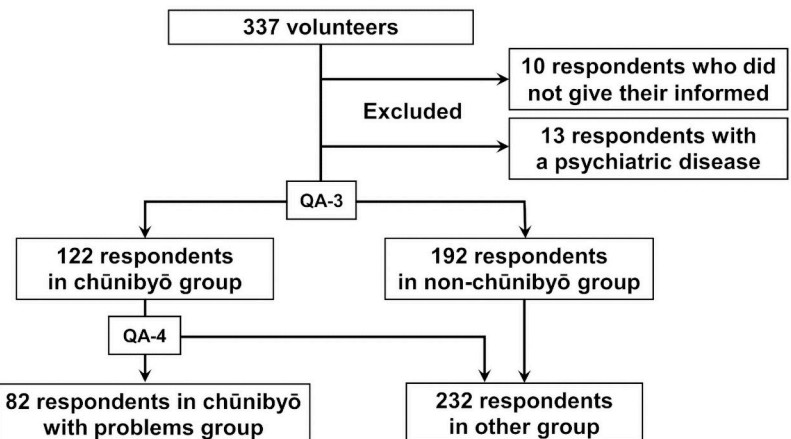

**Fig 1. The flowchart of the study.** QA-3: Did you experience chūnibyō during your middle/high school years? Respondents who replied "Yes" or "I do not think I did, but I was told I did." were classified in the chūnibyō group, and respondents who replied "No" or "I am not sure if I did, and nobody said I did" were classified in the non-chūnibyō group. QA-4: Have you had any problems due to chūnibyō or any other problems related to puberty? Respondents who selected items other than "No nor have I had chūnibyō" were classified in the other group, and the rest were classified in the chūnibyō with problems group.

signatures were not collected because the questionnaire was anonymous. We obtained participant consent through respondents selecting a checkbox indicating their intention to agree in an online questionnaire. This study protocol was approved by the Institutional Review Board of Fukujuji Hospital (Study number: 20064).

## Definition

Respondents were classified in the chūnibyō group if they answered "Yes" or "I do not think I am chūnibyō, but I was told I was chūnibyō" to question (Q) A-3 "Did you experience chūnibyō during your middle/high school years?". Respondents were classified in the non-chūnibyō group if they answered "No" or "I am not sure if I was chūnibyō, and nobody told me that I was" to QA-3. Respondents in the chūnibyō group were classified in the chūnibyō with problems subgroup if they answered yes to the QA-4: "Have you had any problems due to chūnibyō or any other problems related to puberty?". Respondents in the chūnibyō group who replied "No, nor have I experienced chūnibyō" to QA-4 and respondents in the non-chūnibyō group were classified in the other group. QB-3 asked, "Did you call yourself something other than "Boku"/"Ore" if you are male or "Watasi"/"Atasi" if you are female?". Japanese have many types of first-person pronouns. Generally, males call themselves "boku" or"ore", and females call themselves "watasi" or"atasi" [10]. However, some people call themselves different terms, such as their first name, "sessha", "assi", "oira", "soregasi", "boku (for females)", or others to indicate their unique character [2,11].

## Statistical methods

All data were analysed and processed using EZR, version 1.53 [12]. Student's t tests, Mann-Whitney U tests, and Fisher's exact tests were used to compare differences between groups. A receiver operating characteristic (ROC) curve was constructed, and the area under the receiver operating characteristic curve (AUC) was calculated for each predictive model. ROC curves were used to determine the cut-off values. The AUC is an accurate measure of the predictive ability of a model. The level of statistical significance was set at $p = 0.05$ (2-tailed).

**Table 2. Comparisons between the chūnibyō group and the non-chūnibyō group.**

| | Chūnibyō group (n = 122) | Non-"chūnibyō" group (n = 192) | *p*-value |
|---|---|---|---|
| Age, median (IQR), years | 31 (26–37) | 31 (26–40) | 0.740 |
| Sex (male/female) | 79/43 | 96/96 | 0.011 |
| QA-4 Have you had any problems due to chūnibyō or any other problems related to puberty?, n (%) | 82 (67.2) | 27 (14.1) | <0.001 |
| Difficulty communicating, n (%) | 30 (24.6) | 9 (4.7) | <0.001 |
| Deterioration of relationships with family and/or friends, n (%) | 28 (23.0) | 5 (2.6) | <0.001 |
| Stressed with no clear cause, n (%) | 43 (35.2) | 12 (6.3) | <0.001 |
| Picked on for experiencing chūnibyō, n (%) | 3 (2.5) | 0 (0.0) | 0.058 |
| Bullied, n (%) | 10 (8.2) | 3 (1.6) | 0.007 |
| Became socially withdrawn and/or unemployed, n (%) | 2 (1.6) | 0 (0.0) | 0.150 |
| Still experienced chūnibyō in adulthood, n (%) | 21 (17.2) | 0 (0.0) | <0.001 |
| Other, n (%) | 9 (7.4) | 9 (4.7) | 0.330 |
| QA-5 Had any embarrassing experiences, n (%) | 105 (86.1) | 139 (72.4) | 0.632 |
| The memory is bad, n (%) | 60 (49.2) | 66 (34.4) | 0.010 |
| The memory itself is not bad, n (%) | 46 (37.7) | 73 (38.0) | 1.000 |
| QB-1, Agreed that academic tests were not enough to measure their true abilities, n (%) | 58 (47.5) | 66 (34.4) | 0.024 |
| QB-2 Agreed that they felt that there was something wrong with the world, n (%) | 77 (61.1) | 67 (34.9) | <0.001 |
| QB-3 Called themselves something other than "Boku"/"Ore" if male or "Watasi"/"Atasi" if female, n (%) | 34 (27.9) | 46 (24.0) | 0.507 |
| QB-4 Had imaginary/fantasy friends or boyfriends/girlfriends, n (%) | 39 (32.0) | 10 (5.2) | <0.001 |
| QB-5 Confessed their love to others or had others confess their love to them when they were middle/high school students, n (%) | 86 (70.5) | 148 (77.1) | 0.232 |

## Results

### Characteristics of the chūnibyō group: Comparisons with the characteristics of the non-chūnibyō group

Among 314 respondents, 122 respondents (38.9%) were identified as belonging to the chūnibyō group, and 192 respondents (61.1%) were identified as belonging to the non-chūnibyō group (Table 2). The median age was not significantly different between the chūnibyō group and the non-chūnibyō group (median age of 31 years (interquartile range (IQR) 26–37) vs. median 31 years (IQR 26–40), *p* = 0.740). Seventy-nine of the 122 respondents in the chūnibyō group (64.8%) were male, which was a greater proportion than in the non-chūnibyō group (n = 96 (50.0%), *p* = 0.011). There was a significantly greater proportion of respondents in the chūnibyō group who reported having any problems due to chūnibyō or any other problems related to puberty (n = 82 (67.2%) vs. 27 (14.1%), *p*<0.001). In the chūnibyō group, feeling stressed with no clear cause was the most common problem (n = 43, 35.2%), and many respondents also had difficulties communicating (n = 30, 24.6%) and had experienced the deterioration of their relationships with family and/or friends (n = 28, 23.0%). Twenty-one respondents in the chūnibyō group (17.2%) still experienced chūnibyō as adults. Having experienced any embarrassment was not significantly different between the chūnibyō group and the non-chūnibyō group (n = 105 (86.1%) vs. n = 139 (72.4), *p* = 0.507), although a greater proportion of the respondents in the chūnibyō group perceived the embarrassing experience to be a bad memory (n = 60 (49.2%) vs. n = 66 (34.4%), *p* = 0.010). There were significantly more respondents in the chūnibyō group who "thought that academic tests were not enough to measure their true abilities" (QB-1) (n = 58 (47.5%) vs. n = 66 (35.4%), *p* = 0.024), "felt that something was wrong with the world" (QB-2) (n = 77 (61.1%) vs. n = 67 (34.9%),

$p$<0.001), and "had any imaginary/fantasy friends or boyfriends/girlfriends" (QB-4) (n = 39 (32.0%) vs. n = 10 (5.2%), $p$<0.001). Regarding QB-6, which asked, "If someone had told you that you were chūnibyō when you were an adolescent, how would you have reacted?", 63 respondents in the chūnibyō group (51.3%) and 45 respondents in the non-chūnibyō group (23.4%) indicated that they would have accepted that evaluation, 37 respondents in the chūnibyō group (30.3%) and 109 respondents in the non-chūnibyō group (56.8%) indicated that they would have rejected that evaluation, 8 respondents in the chūnibyō group (6.6%) and 13 respondents in the non-chūnibyō group (6.8%) indicated that they would have felt positively about that evaluation, and 37 respondents in the chūnibyō group (30.3%) and 43 respondents in the non-chūnibyō group (22.4%) indicated that would have had a negative reaction.

## Characteristics of the chūnibyō with problems subgroup: Comparisons with the characteristics of the other group

Eighty-two respondents (26.1%) were classified in the chūnibyō with problems subgroup, while 232 respondents were classified in the other group (Table 3). In the chūnibyō with problems subgroup, the median age was 33 years old (IQR 26–38), and there were 51 males (62.2%); the age and sex distributions did not differ between the chūnibyō with problems subgroup and the other group (age: median 31 years old (IQR 26–39), $p = 0.445$, male: n = 124 (53.4%), $p = 0.196$). There were significantly greater proportions of respondents in the chūnibyō with problems subgroup who "thought that academic tests were not enough to measure their true abilities" (QB-1) (n = 43 (52.4%) vs. n = 81 (34.9%), $p = 0.006$), "felt that something was wrong with the world" (QB-2) (n = 61 (74.4%) vs. n = 83 (35.8%), $p$<0.001), and "had any imaginary/fantasy friends or boyfriends/girlfriends" (QB-4) (n = 27 (32.9%) vs. n = 22 (9.5%), $p$<0.001).

## Development of a predictive score to identify chūnibyō

We developed a predictive score to identify chūnibyō, which included the following three variables: thinking that academic tests were not enough to measure their true abilities, feeling that something was wrong with the world, and having any imaginary/fantasy friends or boyfriends/girlfriends. Each variable was assigned a value of 1 point; thus, the maximum total score was 3 points. The ROC of the score is shown in Fig 2. The AUC was 0.721 (95% confidence interval (Cl) 0.658–0.784). When the cut-off value was 2 points or more, the sensitivity, specificity, and odds ratio were 59.8%, 76.3%, and 4.75 (95% Cl 2.70–8.46), respectively.

**Table 3. Comparisons between people who experienced problems due to chūnibyō and others.**

| | Problems due to chūnibyō (n = 82) | Others (n = 232) | $p$-value |
|---|---|---|---|
| Age, median (IQR), years | 33 (26–38) | 31 (26–39) | 0.445 |
| Sex (male/female) | 51/31 | 124/108 | 0.196 |
| QA-5 Had any embarrassing experiences, n (%) | 69 (84.1) | 176 (75.9) | 0.162 |
| QB-1 Agreed that academic tests were not enough to measure their true abilities, n (%) | 43 (52.4) | 81 (34.9) | 0.006 |
| QB-2 Agreed that they felt that there was something wrong with the world, n (%) | 61 (74.4) | 83 (35.8) | <0.001 |
| QB-3 Called themselves something other than "Boku"/"Ore" if male or "Watasi"/"Atasi" if female, n (%) | 21 (25.6) | 59 (25.4) | 1.000 |
| QB-4 Had imaginary/fantasy friends or boyfriends/girlfriends, n (%) | 27 (32.9) | 22 (9.5) | <0.001 |
| QB-5 Confessed their love to others or had others confess their love to them when they were middle/high school students, n (%) | 56 (68.3) | 178 (76.7) | 0.142 |

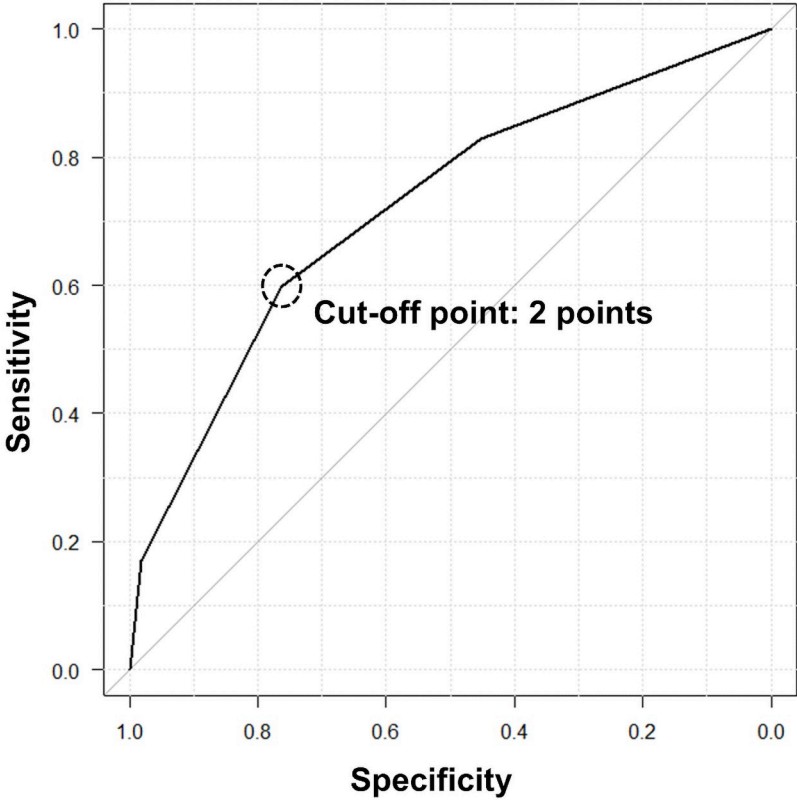

**Fig 2. The ROC of the predictive score for the identification of chūnibyō.** The score included the following three variables: Thinking that academic tests were not enough to measure their true abilities, feeling that something was wrong with the world, and having any imaginary/fantasy friends or boyfriends/girlfriends. Each variable was assigned a value of 1 point; thus, the maximum total score was 3 points. The AUC was 0.721 (95%Cl 0.658–0.784). When the cut-off value was 2 points or more, the sensitivity, specificity, and odds ratio were 59.8%, 76.3%, and 4.75 (95% Cl 2.70–8.46), respectively. ROC: Receiver operating characteristic, AUC: The area under the receiver operating characteristic curve, Cl: Confidence interval.

## Discussion

This study was performed to identify the characteristics of chūnibyō by collecting the experiences and thoughts of people who considered themselves or who others considered to be chūnibyō through an anonymous questionnaire because a standard definition of chūnibyō has not been developed. The chūnibyō group was predominantly male and had experienced more problems. Chūnibyō was characterized by feeling that academic tests did not reflect true worth, feeling uncomfortable in the world, and having imaginary/fantasy friends or boyfriends/girlfriends. Many people in both the chūnibyō and non-chūnibyō groups had negative impressions of chūnibyō. The score, which included feeling that academic tests did not reflect their true worth, feeling uncomfortable in the world, and having imaginary/fantasy friends or boyfriends/girlfriends, was developed to identify people who experience problems due to chūnibyō. The score had a high AUC, and a high sensitivity, specificity, and odds ratio were obtained when the cut-off value was 2 points or more. This knowledge might be useful for identifying children with chūnibyō and helping them establish a better environment.

## Characteristics of chūnibyō

Generally, children in puberty experience stress because of isolation from their parents and teachers, during which time their physical development continues [5,9]. Some children have a high opinion of their own competence based on an underestimation of the value of others and intense self-involvement, and they alternate between high expectations and a poor perception of their identity [9,13]. More than their actual ability or the desire for approval, this high self-evaluation can lead to the development of chūnibyō in children during puberty [5]. Japanese "chū-ni" (eight-grade) students have an abundance of time for self-reflection [5]. In Japan, seventh-grade students have just entered middle school and need to adapt to their new environment, and nine-grade students are in the process of preparing to take high school entrance examinations [5]. However, "chū-ni" students have a relatively large amount of free time because they have already become accustomed to their school, and they have not yet started to prepare for high school entrance examinations in earnest [5]. In our study, many people in the chūnibyō group thought academic tests did not reflect their true worth and felt uncomfortable in the world, which may indicate an imbalance between reality and their self-evaluation. A difference between their scores on academic tests and their ideals could affect their self-esteem; therefore, people with chūnibyō deny the value of academic tests and the world [4,5]. Furthermore, people with chūnibyō sometimes act distinctively and stand out from others [4]. A higher proportion of those in the chūnibyō group reported having imaginary/fantasy friends or boyfriends/girlfriends. They usually know that their imaginary/fantasy friends and boyfriends/girlfriends are fictional and simply use them to set themselves apart from others [4]. Therefore, the chūnibyō behaviour of having imaginary/fantasy friends or boyfriends/girlfriends is different from a delusion due to schizophrenia [14].

Our study did not show a significant difference between the chūnibyō group and the non-chūnibyō group in the personal pronouns used or in their love confessed for another person. Variations in first-person pronouns might seem to be a characteristic of chūnibyō, given the desire to set themselves apart from others [4]; however, the selection of various personal pronouns might be common during puberty. We thought that people with chūnibyō might struggle to confess their love for another person due to their poor communication skills; however, apparently, people with chūnibyō can confess their love. Additionally, having embarrassing experiences was also not significantly different in our study. Generally, it has been considered that having embarrassing experiences, called "Kurorekisi" in Japanese, is common for people with chūnibyō [4,5]. Our study revealed that even people without chūnibyō have embarrassing experiences, although people with chūnibyō tend to perceive those experiences as bad memories. In the chūnibyō group, 40 of the 122 respondents had never had any problems due to chūnibyō. We believe that having any problems due to chūnibyō could be associated with the severity of chūnibyō.

## Chūnibyō is not disease

People with chūnibyō who experience problems are a particularly important subgroup. There has been no report describing which problems affect people with chūnibyō, and our study is the first to report these problems. In our study, approximately two-thirds of the chūnibyō group had experienced any problems related to chūnibyō, including feeling stressed with no clear cause, having poor communication skills, and experiencing the deterioration of relationships. These features seem to be similar to those of autism spectrum disorder [15]. Although chūnibyō cannot be denied to be part of mild autism spectrum disorder, many people experience chūnibyō, and they can live a social life without problems related to chūnibyō when they grow up. Therefore, chūnibyō is not considered a psychiatric disorder [2,4,7], and it might be

considered to distinguish between chūnibyō and autism spectrum disorder. Furthermore, chūnibyō is not characterized by neurodevelopmental disorders, episodes of self-harm, the deterioration of motor skills, attention deficits or careless mistakes; therefore, it is different from other psychiatric disorders, such as attention deficit hyperactivity disorder, personality disorder, Rett syndrome, and others [16–18]. If chūnibyō can be detected, it might help adults, such as parents and teachers, establish a better environment for the children. Therefore, it is important to identify the characteristics of chūnibyō because children who experience chūnibyō sometimes struggle to accept that fact. Indeed, approximately one-third of the respondents in the chūnibyō group responded to the question "If someone had told you that you were chūnibyō when you were an adolescent, how would you have reacted?" that they would have rejected the statement. Therefore, a predictive score that could be used to identify adolescents who were likely to experience problems associated with chūnibyō was developed. The score might also be useful to identify people who are likely to remain chūnibyō after puberty, although chūnibyō is generally resolved after puberty or school [5]. According to one theory, it is not that chūnibyō resolves but rather that adults are successful at hiding their excessive self-consciousness [5]. An awareness of chūnibyō on the part of those affected by it might be an important step in the resolution of problems associated with chūnibyō. However, caution should be exerted owing to the negative impression of chūnibyō that many people affected by it have. We believe that adults who interact with children with chūnibyō who experience any problems should notice and accept the mental changes in these children and indirectly help the children to become aware of their chūnibyō. We also believe that no intervention is necessary when children with chūnibyō do not experience any associated problems. Importantly, chūnibyō is not a disease but a transient expression of individuality, and ridicule should be avoided.

## Chūnibyō in other countries

The unique behaviours that are characteristic of chūnibyō, such as the application of a bandage to "seal in" the power granted them by a devil, insisting that they can see a supernatural being, and calling themselves special names, seem to refer to anime, manga, cosplay, and gaming. Chūnibyō might be related to aspects of Japanese culture [1,4,5]. However, we do not believe that Japanese culture adversely affects children during puberty, although there are dissenting opinions [19,20], and we do not believe that people with chūnibyō are abnormal or harmful. Because chūnibyō is thought to be based on high self-evaluation and the desire for approval during puberty [3–5], even without the effect of the aforementioned forms of media, children with chūnibyō may appear as a different phenotype. Furthermore, the culture and the social climate generally influence each other [21]; hence, many forms of media such as anime, manga, cosplay, and gaming may conversely refer to people with chūnibyō. Recently, Japanese culture has started to spread globally [22], and chūnibyō may become a common problem in many countries other than Japan in the future. While the existence of ethnic differences in chūnibyō is unclear, the environment also influences adolescents through their experiential realities, which impact their well-being and mental health [23]. Therefore, if environments in the world are influenced by the spread of Japanese culture, people with chūnibyō might be identified in other countries in the near future. However, it is an assumption without any evidence. Japanese culture has spread to various parts of the world, and many children and teenagers are addicted to Japanese games or anime. Therefore, negative impacts that are similar to the symptoms of Chūnibyō may occur.

## Limitations

This investigation had several limitations. It was conducted as an online survey, and respondents were recruited on a blog, SNSs, and internet bulletin boards. The definition of chūnibyō was based on either a self-report or a previous statement made by others; therefore, some of the respondents in the non-chūnibyō group could have actually belonged to the chūnibyō group. Furthermore, all respondents were adults aged 20 years or older, and we identified respondents who had experienced chūnibyō in the past. We believed that children could not be evaluated with regard to chūnibyō because they might not be aware of experiencing chūnibyō [4] or they might reject their own experience of chūnibyō, as shown in this study.

## Conclusion

The study is the first to report the characteristics of chūnibyō by collecting the experiences and thoughts of people considered to be chūnibyō. Adults who interact with children with chūnibyō may be able to support these children's mental health if they are aware of the characteristics of chūnibyō.

## Supporting information

**S1 Data.**
(XLSX)

## Author Contributions

**Conceptualization:** Masafumi Shimoda.

**Data curation:** Masafumi Shimoda, Kozo Morimoto, Yoshiaki Tanaka, Kozo Yoshimori.

**Formal analysis:** Masafumi Shimoda.

**Investigation:** Masafumi Shimoda.

**Methodology:** Masafumi Shimoda, Kozo Morimoto.

**Project administration:** Ken Ohta.

**Software:** Masafumi Shimoda.

**Supervision:** Masafumi Shimoda, Kozo Morimoto, Ken Ohta.

**Validation:** Masafumi Shimoda.

**Visualization:** Masafumi Shimoda.

**Writing – original draft:** Masafumi Shimoda.

**Writing – review & editing:** Masafumi Shimoda, Kozo Morimoto.

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
