## [Decision Letter · Decision Letter 0]

29 Oct 2021

PONE-D-21-21951Characteristics of “chūnibyō” identified by a questionnairePLOS ONE

Dear Dr. Shimoda,

Thank you for submitting your manuscript to PLOS ONE. After careful consideration, we feel that it has merit but does not fully meet PLOS ONE’s publication criteria as it currently stands. Therefore, we invite you to submit a revised version of the manuscript that addresses the points raised during the review process.

I agree that it it worth considering the effects across cultures where it is known by different names. It is similar to the Messiah Complex that occurs in Christian societies and may also be related to the prophetic ideal.

We look forward to receiving your revised manuscript.

Kind regards,

Andrew R. Dalby, PhD

Academic Editor

PLOS ONE

2. Please provide additional details regarding participant consent. In the Methods section, please ensure that you have specified (1) whether consent was informed and (2) what type you obtained (for instance, written or verbal). If your study included minors, state whether you obtained consent from parents or guardians. If the need for consent was waived by the ethics committee, please include this information.

5. Please upload a copy of Figure 1 of page 11 and 2 for page 21, to which you refer in your text. If the figure is no longer to be included as part of the submission please remove all reference to it within the text.

Reviewers' comments:

Reviewer's Responses to Questions

**Comments to the Author**

1. Is the manuscript technically sound, and do the data support the conclusions?

Reviewer #1: Yes

2. Has the statistical analysis been performed appropriately and rigorously? 

Reviewer #1: Yes

3. Have the authors made all data underlying the findings in their manuscript fully available?

Reviewer #1: Yes

4. Is the manuscript presented in an intelligible fashion and written in standard English?

Reviewer #1: Yes

5. Review Comments to the Author

Reviewer #1: This article is very interesting and innovative. This fulfills its promise of demonstrating the characteristics of chūnibyō, something scholars have never comprehensively formulated. By using a mixed approach between quantitative and qualitative, this article is able to innovatively formulate the characteristics of chūnibyō. The working method is explained in a sequential manner and written in a clear and logical manner so that the results are very convincing. The results of this study will be an important reference for future research on similar topics. In addition, this article will provide inspiration to conduct research in the same field in the future (see below).

As an original and innovative study on chunibyo, this article deserves to be published.

Minor suggestions are only given to the “Chūnibyō in other countries” subsection. This section is written with assumptions, without evidence and references, but the points presented are quite important and interesting. For this reason, it is recommended that this point be placed at the end of the article as a future topic research. It is a matter of facts, that Japanese culture has spread to various parts of the world, many children and teenagers are addicted to Japanese games or anime, and this means that negative impacts that are similar to the symptoms of Chūnibyō, may occur. If true, this is a good research topic because of its cross-country, cross-cultural, and cross-disciplinary characteristics.

6. PLOS authors have the option to publish the peer review history of their article (what does this mean?). If published, this will include your full peer review and any attached files.

Reviewer #1: **Yes: **Dr. Ida Ayu Laksmita Sari

---

## [Author Response · Author response to Decision Letter 0]

3 Nov 2021

Responses to the editor and the reviewers

PONE-D-21-21951

“Characteristics of “chūnibyō” identified by a questionnaire”

All modifications have been highlighted in the revised manuscript.

To the editor and the reviewer:

Thank you very much for your constructive comments on our manuscript PONE-D-21-21951. We are pleased to hear that the reviewer found our study intriguing. In response to the reviewer’s requests and comments, we generated text. We carefully studied all comments and made the necessary edits/modifications. The revised version of the manuscript has been edited by a professional English-editing service. The point-by-point responses are listed below. We hope that we have sufficiently addressed the issues raised by the reviewer.

Regarding the editor's comments:

I agree that it it worth considering the effects across cultures where it is known by different names. It is similar to the Messiah Complex that occurs in Christian societies and may also be related to the prophetic ideal.

Response: Thank you very much for your comments and important introductions. It is very interesting, and we think that Messiah Complex is similar to Chūnibyō. However, Chūnibyō can generally occur during puberty. On the other hand, Messiah Complex may be not related to puberty. Therefore, we did not include Messiah Complex in our report.

Regarding the reviewer's comments:

Reviewer #1: This article is very interesting and innovative. This fulfills its promise of demonstrating the characteristics of chūnibyō, something scholars have never comprehensively formulated. By using a mixed approach between quantitative and qualitative, this article is able to innovatively formulate the characteristics of chūnibyō. The working method is explained in a sequential manner and written in a clear and logical manner so that the results are very convincing. The results of this study will be an important reference for future research on similar topics. In addition, this article will provide inspiration to conduct research in the same field in the future (see below).

Response: Thank you very much for your comments. It is very encouraging. We are glad you reviewed our manuscript. We modified the text based on your comment.

-Minor suggestions are only given to the “Chūnibyō in other countries” subsection. This section is written with assumptions, without evidence and references, but the points presented are quite important and interesting. For this reason, it is recommended that this point be placed at the end of the article as a future topic research. It is a matter of facts, that Japanese culture has spread to various parts of the world, many children and teenagers are addicted to Japanese games or anime, and this means that negative impacts that are similar to the symptoms of Chūnibyō, may occur. If true, this is a good research topic because of its cross-country, cross-cultural, and cross-disciplinary characteristics.

Response: Thank you very much for your helpful comments and important introductions. It’s exactly as your comment, so we modified it on page 28, lines 357-361.

Response: We confirmed that our manuscript met the PLOS ONE's style requirements.

2. Please provide additional details regarding participant consent. In the Methods section, please ensure that you have specified (1) whether consent was informed and (2) what type you obtained (for instance, written or verbal). If your study included minors, state whether you obtained consent from parents or guardians. If the need for consent was waived by the ethics committee, please include this information.

Response: Our manuscript demonstrated participant consent. Informed consent was obtained from all respondents in an anonymous online questionnaire, although signatures were not collected because the questionnaire was anonymous. We obtained participant consent through respondents selecting a checkbox indicating their intention to agree in an online questionnaire. This study protocol was approved by the Institutional Review Board of Fukujuji Hospital (Study number: 20064). We added type of an obtention for participant consent in our manuscript on page 7, lines 105-107.

Response: We uploaded data availability statement of our study as supplemental file newly. Please confirm it.

Response: Same as response to Q3, we uploaded data availability statement of our study as supplemental file newly. Please confirm it.

5. Please upload a copy of Figure 1 of page 11 and 2 for page 21, to which you refer in your text. If the figure is no longer to be included as part of the submission please remove all reference to it within the text.

Response: We uploaded them on the submission system.

---

## [Editor Report · Decision Letter 1]

9 Nov 2021

Characteristics of “chūnibyō” identified by a questionnaire

PONE-D-21-21951R1

Dear Dr. Shimoda,

We’re pleased to inform you that your manuscript has been judged scientifically suitable for publication and will be formally accepted for publication once it meets all outstanding technical requirements.

Kind regards,

Andrew R. Dalby, PhD

Academic Editor

PLOS ONE
---

## [Editor Report · Acceptance letter]

12 Nov 2021

PONE-D-21-21951R1 

Characteristics of “chūnibyō” identified by a questionnaire 

Dear Dr. Shimoda:

I'm pleased to inform you that your manuscript has been deemed suitable for publication in PLOS ONE. Congratulations! Your manuscript is now with our production department. 

Kind regards, 

on behalf of

Dr. Andrew R. Dalby 

Academic Editor

PLOS ONE